# Comparative Evaluation of Adsorption of Major Enzymes in a Cellulase Cocktail Obtained from *Trichoderma reesei* onto Different Types of Lignin

**DOI:** 10.3390/polym14010167

**Published:** 2022-01-01

**Authors:** Dae-Seok Lee, Younho Song, Yoon-Gyo Lee, Hyeun-Jong Bae

**Affiliations:** 1Bio-Energy Research Center, Chonnam National University, Gwangju 500-575, Korea; realstone@hanmail.net (D.-S.L.); jagnad@naver.com (Y.S.); 2Department of Wood Science and Landscape Architecture, Chonnam National University, Gwangju 500-757, Korea; spake123@naver.com; 3Department of Bioenergy Science and Technology, College of Agriculture and Life Sciences, Chonnam National University, Gwangju 500-757, Korea

**Keywords:** lignin adsorption, cellobiohydrolases (CBHs), endoglucanase, β-glucosidase, xylanase, mannanase

## Abstract

Cellulase adsorption onto lignin decreases the productivity of enzymatic hydrolysis of lignocellulosic biomass. Here, adsorption of enzymes onto different types of lignin was investigated, and the five major enzymes—cellobiohydrolases (CBHs), endoglucanase (Cel7B), β-glucosidase (Cel3A), xylanase (XYNIV), and mannanase (Man5A)—in a cellulase cocktail obtained from *Trichoderma reesei* were individually analyzed through SDS-PAGE and zymogram assay. Lignin was isolated from woody (oak and pine lignin) and herbaceous (rice straw and kenaf lignin) plants. The relative adsorption of CBHs compared to the control was in the range of 14.15–18.61%. The carbohydrate binding motif (CBM) of the CBHs contributed to higher adsorption levels in oak and kenaf lignin, compared to those in pine and rice lignin. The adsorption of endoglucanase (Cel7B) by herbaceous plant lignin was two times higher than that of woody lignin, whereas XYNIV showed the opposite pattern. β-glucosidase (Cel3A) displayed the highest and lowest adsorption ratios on rice straw and kenaf lignin, respectively. Mannanase (Man5A) was found to have the lowest adsorption ratio on pine lignin. Our results showed that the hydrophobic properties of CBM and the enzyme structures are key factors in adsorption onto lignin, whereas the properties of specific lignin types indirectly affect adsorption.

## 1. Introduction

Lignocellulosic biomass is considered an alternative to petroleum as an energy resource and is used to obtain bioconversion products for subsequent biochemical and/or energy production [1]. However, the structural complexity of lignocellulose and its recalcitrance to degradation by hydrolytic enzymes reduces saccharification efficiency, and increase the costs to improve bioconversion rate, which, in turn, reduces the price competitiveness of the bioconversion products.

The presence of lignin hinders enzymatic hydrolysis of lignocellulose. Two major factors contribute to the negative impact of lignin on hydrolysis. The first factor is the structural recalcitrance of lignocellulose due to lignin. Lignocellulose is composed of three major polymers: lignin, hemicellulose, and cellulose. Lignin is a highly oxygenated aromatic polymer and binds cellulose microfibrils comprised of β-1,4-D-glucose polysaccharide chain bundles, ranging from 10 to 35 nm in diameter, together with a hemicellulose linker. Cellulose is tightly packed with inter- and intra- hydrogen bonds between individual cellulose chains. The difference in hydrogen-bonding networks between each cellulose chain within the microfibril unit results in three different recalcitrance behaviors [2,3]. During pretreatment of lignocellulosic biomass, lignin acts as a physical and chemical barrier to restrict cellulose swelling or structural modification of cellulose. Lee et al. (2020) showed that the lignin barrier affects cellulose modification under different pretreatment conditions, and three types of cellulose are released based on the enzymatic hydrolysis rate [4]. This is due to differences in lignin composition and compatibility between the pretreatment method and the lignocellulose feedstock type. The second factor is an enzyme-related retardation of hydrolysis. Cellulolytic and xylanolytic enzymes have been reported to be adsorbed onto lignin, and thereby accessibility of these hydrolytic enzymes is restricted. In particular, cellobiohydrolases (CBHs) with a tunnel-shaped catalytic site and hydrophobic carbohydrate binding motif (CBM) interact preferentially with hydrophobic surfaces of solid cellulose fibers to release soluble cellobiose through the progressive action [5,6,7]. Lignin also associates preferentially with the hydrophobic surfaces of cellulose and cellulase (steric hindrance), and was previously reported to block enzyme-cellulose productive adsorption [8,9]. Thus, significant inhibition of enzymatic hydrolysis occurs in lignin-containing pretreated lignocellulose. 

Pretreatment of the lignocellulosic biomass is required to remove or reduce inhibition by lignin, and to modify cellulose structure to enhance enzymatic hydrolysis. Kumar et al. (2009) performed a characterization of cellulase adsorption capacities of lignin and cellulose in biomass pretreated with different methods such as ammonia fiber expansion (AFEX), ammonia recycle percolation (ARP), dilute sulfuric acid (DA), flowthrough (FT), lime, and steam explosion with SO_2_ [10]. The isotherm parameters of enzyme adsorption have been summarized based on diverse types of lignin isolated from different pretreated-lignocellulose samples. The lignin isolated from pine, poplar and corn stove pretreated by organosolv or steam explosion showed a much higher maximum adsorption capacity than that pretreated by alkali, dilute acid, ammonia, and sulfonated alkali [11]. Lignin in organosolv pretreated pine was characterized with high hydrophobicity, and a 23–30% reduction in hydrophobicity of the lignin by carboxylation and sulfonation led to a 76–96% reduction in lignin inhibition [12]. These results clearly indicate that non-productive lignin adsorption on the cellulase negatively affects the enzymatic hydrolysis rate.

Enzyme adsorption onto lignin has been reported with chemically modified [12,13,14,15,16,17] and native lignin [18]. However, quantification of the enzyme adsorption has been performed with a single type of enzyme, which was purified from the cellulase cocktail or produced via recombinant expression. In these cases, a single type of enzyme is clearly over-adsorbed, compared to the case when a cellulase cocktail is used. Therefore, quantification of adsorption of specific types of enzymes onto native lignin in a cellulase cocktail is also necessary. Such a quantification study will provide a standard for the adsorption capacity of specific enzyme types that can be altered depending on different types of lignin. To this end, lignin isolated from the lignocelluloses pretreated with popping methods where no chemicals are used [19,20] may be an appropriate material to serve as the standard for quantification of adsorption capacity. Here, we isolated lignin from various lignocellulosic biomass types pretreated with popping methods, including hardwood (*Quercus acutissima*), softwood (*Pinus densiflora*), and agricultural herbaceous plants (rice straw and kenaf), and analyzed the adsorption ratio of the major enzymes that function as gate keepers for the cleavage of β-1,4 glycosidic linkages in a cellulase cocktail from *Trichoderma reesei*. Other enzymes, such as β-glucosidase from *Aspergillus niger* and xylanase from *Thermomyces lanuginosus*, were also quantified as supplementary enzymes with the native lignin.

## 2. Materials and Methods

### 2.1. Enzyme Preparation

The cellulase cocktail (*Trichoderma reesei*, celluclast 1.5 L) were purchased from Novozymes (Bagsvᴂrd, Denmark). β-glucosidase (*Aspergillus niger*) were purchased from Megazyme (Lot 141001, Wicklow, Ireland). The xylanases from *Thermomyces lanuginosus* were purchased from Sigma-Aldrich (X2753-50G, St. Louis, MI, USA).

### 2.2. Lignin Isolation

Hardwood (oak, *Quercus acutissima*), softwood (pine, *Pinus densiflora*), and agricultural herbaceous plants (rice straw and kenaf) were chopped into lengths of approximately 2 cm, and soaked in tap water for 1 day before placement in a laboratory-scale cast iron cylindrical reactor (3 L) to conduct the popping pretreatment [19]. The reactor was heated at a rate of between 15 and 20 °C per minute until 220 °C and 298.69 Pa (21 kg f cm^−2^). The hatch was rapidly opened to expose the sample to atmospheric pressure. The popped samples were dried and ground to 251–422 μm particle size with a Willy mill fitted equipped with stainless steel blades. Lignin isolation from the samples was conducted with a fresh cellulase cocktail in all reactions that repeated until the carbohydrates were nearly removed. The isolated lignin were filtered through a 100 mesh screen cup (Cot. S3895, Sigma-Aldrich, St. Louis, MI, USA), freeze dried, and stored at room temperature.

### 2.3. Lignin Adsorption

Enzyme adsorption onto lignin was performed in 1 mL of 20 mM citrate buffer (pH 5.0) with 0.5, 1.0, 2.5, 5.0, 10.0, 15.0, and 20.0 mg mL^−1^ lignin (oak, pine, rice straw, and kenaf) and 15 FPU cellulase cocktail mL^−1^ (126 μg·mL^−1^) at 50 °C for 1 h. The tubes were centrifuged to separate supernatant and pellet at 35,000 RCF (relative centrifugal force; CT15RE, Hitachi Koki Co., Ltd., Tokyo, Japan) for 10 min. The supernatants were diluted 5–10 times with 20 mM citrate buffer (pH 5.0) for consistent and accurate protein quantification. The free protein in each supernatant was measured with Bio-Rad protein assay solution (Cat. No. 500-0006, BIO-RAD, Hercules, CA, USA) and ELISA (MULTISCAN EX, Thermo Scientific, Waltham, MI, USA) at 595 nm. The protein concentration adsorbed onto the lignin was quantified using the following equation:
[E]_ad_ = [E]_total_ − [E]_free_(1)
where [E]_ad_ is the protein concentration adsorbed onto the lignin. [E]_total_ is the initial enzyme amount before the adsorption. [E]_free_ is the protein concentration in the supernatant after lignin adsorption. The protein standard curve was derived with bovine serum albumin (BSA) and used to calculate the protein concentration in all experiments.

### 2.4. Adsorption of Cellobiohydrolases (CBHs) onto Lignin

The pellet obtained after centrifugation in the previous section was washed two times with 20 mM citrate buffer. After centrifugation at 35,000 RCF for 10 min, the pellet was incubated with dissociation buffer (100 μL 10% SDS, 100 μL sample buffer, and adjusting the final volume to 300 μL with distilled water) to dissociate the enzymes from the lignin. The contents of the tubes were brought to the boil, maintained for 10 min, and then centrifuged at 35,000 RCF for 10 min. The dissociated proteins were then subjected to SDS-PAGE. The control sample was prepared with 20 μL for 15 FPU mL^−1^ cellulase cocktail and brought to its final volume of 300 μL with the dissociation buffer. The loading volumes were 2.5, 5, 10, and 15 μL for the control, and 5, 10, 20, 30, and 40 μL for the dissociation samples from the lignin to produce more bands having measurable and distinguishable intensities. The proteins separated on the electrophoresis gel were stained with Coomassie brilliant blue R-250. Adsorption rate was calculated by measuring the intensities of the bands on the gel with histogram quantification software (Adobe photoshop CS6, Adobe Inc., San Jose, CA, USA) following the method outlined at https://support.dalton.missouri.edu/index.php/wiki/Public:Quantifying_Color_Intensity (accessed on 1 December 2021) [21].

### 2.5. Adsorption of Endoglucanases (EGs) onto Lignin

Adsorption of endo-glucanases (EGs) onto lignin was conducted with 20 μL of the 15 FPU cellulase cocktail (126 μg·mL^−1^) and 10 mg mL^−1^ lignin (oak, pine, rice straw, and kenaf) in 20 mM citrate buffer with a final volume of 1 mL, which was incubated at 50 °C for 1 h. The mixture was then centrifuged at 35,000 RCF for 10 min to obtain the supernatant and pellet fractions. The supernatant and pellet fractions were used to measure the total activity of the EGs and the specific individual enzyme (Cel7B) for the adsorption capacity of the lignin. The control was prepared with a 20 μL cellulase cocktail (15 FPU mL^−1^) and brought to its final volume of 1 mL final volume with 20 mM citrate buffer (pH 5.0). The supernatant (100 μL) including free enzymes was used to measure the EGs activity in 500 μL total volume of 20 mM citrate buffer (pH 5.0) with 100 μL 1% CMC at 50 °C for 30 min.

Zymogram analysis was conducted with a pellet fraction obtained as described above to measure the activity of the endoglucanase Cel7B in the cellulase cocktail. The pellet fractions obtained from oak, pine, rice straw, and kenaf lignin were incubated with 300 μL dissociation buffer at room temperature, and boiled for 10 min. After centrifugation at 35,000 RCF for 10 min, the solution including dissociated enzymes was loaded on SDS-PAGE including 30 μL mL^−1^ 1% CMC. The loading volumes were 1, 2, and 4 μL for the control (10× dilution), and 2, 4, and 8 μL for the dissociated EGs. After electrophoresis, the gel was added into 50 mL refolding buffer containing 50 mL 20 mM citrate buffer (pH 5.0), 10 μL 10 mM CaCl_2_, and 50 μL 10% triton X-100 at room temperature for 40 min. The gel was incubated in 50 mL fresh solution of 20 mM citrate buffer (pH 5.0) at 40 °C for 1 h and washed with 20 mM phosphate buffer (pH 7.0). Then, the gel was stained with 0.1% Congo red solution for 30 min, and destained with 1 M NaCl until bands of hydrolysis zones appeared. For better visual clarity, 0.5% acetic acid was added to the gel to change the background color from red to dark blue. The bands’ intensities were quantified using the same method described above.

### 2.6. Lignin Adsorption on the β-Glucosidases from Different Fungi

β-glucosidase adsorption onto lignin was performed in 20 mM citrate buffer (pH 5.0) with a 1 mL final volume and 20 μL cellulase cocktail (15 FPU mL^−1^) with 10 mg mL^−1^ lignin at 50 °C for 1 h. The mixture was then centrifuged to obtain supernatant and pellet fractions. A total of 100 μL of the supernatant and control solution (described above) were separately incubated in 1 mL of 20 mM citrate buffer (pH 5.0) containing 20 μL of 50 mg mL^−1^ *p*NPG (4-Nitrophenyl β-D-glucopyranoside) at 50 °C for 10 min or with 2 mg mL^−1^ cellobiose at 50 °C for 30 min. The *p*NP standard curve was used to measure the β-glucosidase activity. The products from cellobiose were analyzed by HPLC. 

Adsorption of β-glucosidase from *A. niger* onto lignin was conducted in 1 mL of 20 mM citrate buffer (pH 5.0) containing 25 μg β-glucosidase from *A. niger*, and 10 mg mL^−1^ lignin or without lignin as the control. The tubes were incubated at 50 °C for 1 h. The pellet fractions were separated, and the enzymes were dissociated with 200 μL of final volume of dissociation buffer. The solution including dissociated enzymes and the control were subjected to SDS-PAGE. Sample volumes were 10, 20, and 30 μL, whereas volumes of control were 10, 15, and 20 μL.

### 2.7. Lignin Adsorption on the Xylanases under Different Enzyme Conditions

Adsorption of xylanases from *T. reesei* and *T. lanuginosus* was assessed in 1 mL of 20 mM citrate buffer (pH 5.0) containing 10 mg mL^−1^ lignin and 20 μL of the cellulase cocktail for 15 FPU mL^−1^ or 25 μL of 0.1 g mL^−1^ xylanase of *T. lanuginosus* at 50 °C for 1 h. The supernatant and pellet were separated by centrifugation. Xylanase activities of the supernatants were measured with 100 μL of 2% soluble beechwood xylan in 1 mL of 20 mM citrate buffer at 50 °C for 30 min. The xylanases in the pellet fractions of two fungal species were dissociated with 300 μL of dissociation buffer. The control was prepared with 20 μL of the cellulase cocktail of *T. reesei* or 25 μL xylanase solution of *T. lanuginosus* in the dissociation buffer (100 μL 10% SDS, 100 μL protein sample buffer, and adjusting to 300 μL final volume with distilled water). The samples were loaded on the SDS-PAGE containing 30 μL mL^−1^ of 2% soluble beechwood xylan. The volumes were 1, 2.5, and 5 μL for the control (1/2 diluted), and 5, 10, and 20 μL for the lignin adsorption samples for *T. reesei* xylanase. For the xylanase of *T. lanuginosus*, the volumes were 2.5, 5, and 10 μL for the control (1/10 diluted), and 2.5, 5, and 10 μL for the adsorption samples. Zymogram analysis of samples was performed following the procedure described above.

### 2.8. Adsorption of Mannanase in the Cellulase Cocktail Obtained from T. reesei

Analysis of adsorption of mannanase onto lignin was conducted using the zymogram procedure. The procedure for the preparation of the samples and control was the same as that used for EGs and xylanases. The SDS-PAGE was performed with 50 μL mL^−1^ of 0.5 % glucomannan. The loading volumes are 5, 10, and 20 μL for the control (1/10 diluted), and 20, 30, and 40 μL for the samples. The subsequent experiment steps were as described above for EGs and xylanase.

## 3. Result and Discussion

### 3.1. Adsorption of Extracellular Enzymes of T. reesei onto Lignin

Lignin is synthesized through the radical coupling of the monolignols (*p*-coumaryl, coniferyl, and sinapyl alcohol), which results in generation of three subunits termed *p*- hydroxyphenyl (H), guaiacyl (G), and syringyl (S). These subunits are then randomly incorporated into the lignin polymer [22]. Table 1 shows the composition ratio (%) of the three subunits of lignin in different types of lignocelluloses. The lignin of poplar is composed of 61.9% syringyl (S), 37.8% guaiacyl (G), 0.3% p-hydroxyphenyl (H) subunits, showing a 1.64 S/G ratio [23]. Pine lignin contains 1.7% H, 98.3% G, and 0% S subunits. Corn and *Arabidopsis* lignin are composed of 58.9/38.3% S and 20.1/77.1% G units, respectively. Both lignin types include an H subunit at a 2.8% rate. Accordingly, these two lignin types have S/G ratios of 1.54 and 0.26, respectively. Sewalt et al. (1997) reported that reduction in enzyme activity on pine, poplar, and mixed hardwood lignin is induced by high rates of free phenolic hydroxyl groups, high molecular weight, and high methoxy group content compared to barley straw lignin [24]. Corn stove lignin with high G unit content also allows adsorption of significantly higher levels of CBH1 and xylanase compared to the lignin of the herbaceous plant kenaf, *Arabidopsis*, and its ferulate-5-hydroxylase mutant. The lignin of pine softwood consists of a 95% G subunit and also adsorbs xylanase and CBH1 at remarkable rates (45% and 35%, respectively) [18].

The types of lignin, enzymes, and the choice of pre-treatment method affect lignin–enzyme interactions. The lignin from diluted acid pretreated creeping wild ryegrass was shown to yield the highest level of adsorption of cellulase (Celluclast 1.5 L) and β-glucosidase (Novozyme 188), followed by the lignin from liquid hot water-pretreated mixed hardwood chip with a cellulase cocktail (Cellic Ctec 2, *T. reesei*), the lignin from diluted acid or steam explosion-pretreated corn stove or rice straw with cellulase (accellerase 1000, *T. reesei*), sulfite-pretreated lodgepole pine lignin with cellulase (Celluclast 1.5L), and organosolv-pretreated lodgepole pine lignin with cellulase (Celluclast 1.5L, *T. reesei*) [25]. Organosolv pretreatment produces a relatively pure, unaltered, and high-quality lignin with low molecular weight, and thereby leads to lower levels of lignin–enzyme interaction [26]. 

Here, lignin of oak, pine, rice straw, and kenaf biomass were isolated using popping pretreatment, milled, and subjected to enzymatic hydrolysis, without any chemical modification of the lignin structure. Lignocelluloses of hardwood, softwood, and herbaceous plants are composed of 15–35% lignin, 32–55% cellulose, 15–40% hemicellulose [27]. Hardwood lignin is composed of 25–50%, 0–8%, and 46–75% G, H, and S subunits, respectively [28]. Pine lignin contains 1.7% H, 98.3% G, and no S subunit [23]. The lignin of rice straw was found to be composed of 71%, 5%, and 24% G, H, and S subunits, respectively. Furthermore, β-*O*-4’ alky-aryl ethers are present at 78%, and 10–12% of the linkage dimers are acylated [29]. The kenaf lignin is composed of 40.9%, 1.0%, and 58.1% G, H, and S subunits, respectively [30]. Guo et al. (2014) previously suggested that the presence of a phenolic hydroxyl group affects lignin–enzyme interaction in corn stove and kenaf lignocellulose, and the S/G ratios affects lignin adsorption on cellulase when *Arabidopsis* and its ferulate-5-hydroxylase mutant were compared, except for pine lignin, which is intrinsically composed of high rates of G subunits [18].

**Table 1 polymers-14-00167-t001:** Chemical composition of lignin subunits in different types of plants.

Biomass	Chemical Composition (%)				Ref.
	Guaiacyl Unit (G)	*p*-hydroxyphenyl Unit (H)	Syringyl Unit (S)	S/G Ratio	
Hardwood	25–50	0–8	46–75	1.50–1.84	[28]
*Populus tremuloides*	37.8	0.3	61.9	1.64	[23]
*Quercus suber*	44	1	55	1.2	[31]
*Eucalyptus globulus*	10	1	39	3.8	[31]
Softwood	>95	<5	0	-	[28]
*Pinus daeda*	98.3	1.7	0–1.3	-	[23]
Herbaceous plants					
Rice straw	71	5	22	0.31	[29]
Kenaf	40.9	1.0	58.1	1.40	[30]
Corn stove	42.2	44.9	12.9	0.31	[32]
Arabidopsis	77.1	2.8	20.1	0.26	[23]

We also found that oak lignin showed higher adsorption capacity than other lignin types, with 40% adsorption capacity at 20 mg mL^−1^ lignin concentration (Figure 1). Rice straw lignin showed adsorption of 11–28% of cellulase, and thereby showed the lowest adsorption capacity. The adsorption capacities of the lignin decreased in the following order: oak > pine ≥ kenaf > rice. Hardwood oak lignin showed 5% more adsorption than softwood pine, and rice straw lignin with a 71% G subunit showed the lowest rates of cellulase adsorption. These results are in contrast to previous findings, where pine lignin was reported to show 20% higher enzyme adsorption than aspen lignin [18], yet in agreement with findings of Sewalt et al. (1997) [24]. This indicates that many other factors (e.g., physiochemical properties of lignin and enzymes, and the reaction conditions) and interdependencies between these factors affect adsorption of cellulase enzymes onto lignin [33]. Hence, CBHs, Cel7B, Cel3A, XYLIV, and Man5A in a cellulase cocktail of *T. reesei* were separately investigated, as presented in the following sections, to determine the interactions of woody and herbaceous lignin with a single type of enzyme.

### 3.2. Lignin Adsorption on Cellobiohydrolases (CBHs) and Individual Extracellular Enzyme of T. reesei 

Cellulolytic enzymes produced by *T. reesei* are composed of two cellobiohydrolases (CBHs: Cel7A and Cel6A), six endoglucanases (EGs), and small amounts of other enzymes such as β-glucosidase, six xylanase types (XYLI~VI), β-xylosidase, xyloglucanase, and mannanase [34,35,36,37]. The CBH_f_ (full length of cellobiohydrolases) consists of a catalytic domain (CD) and a carbohydrate binding motif (CBM) and cleaves off cellobiose units from the reducing and non-reducing ends. The CBH_f_ and CD account for 68–78% of the total secretome of *T. reesei* [36]. We observed adsorption of CBHs and CD onto lignin on SDS-PAGE gels stained with Coomassie brilliant blue R-250 (Figure 2). The band corresponding to a molecular weight of 55 kDa indicates CBH_f_, whereas CD is observed on the band corresponding to a molecular weight of 48 kDa, accounting for 30.76% of the total CBH content (including CBH_f_ and CD). Adsorptions of CBH_f_ onto lignin were found to be approximately 12–15% with oak and kenaf, and 7–8% with pine and rice lignin. The adsorption affinities decreased in the following order: oak > kenaf > rice > pine lignin. CD was found to be adsorbed at a rate approximately 2–5 times higher in pine and rice than in oak and kenaf lignin. The adsorption affinities of CD onto the lignin were found to be in the following decreasing order: pine > rice > oak > kenaf lignin. This is consistent with the G unit ratios of these lignin types.

Hydrophobic interactions between enzyme and lignin have also been identified as a major driving force of enzyme adsorption onto lignin [9,38,39]. CBM1s (CBM family 1) of Cel7A and Cel6A have a flat hydrophobic surface that interacts with the hydrophobic surface of the crystalline cellulose [40]. Based on the results of chemical shift changes of amino acids (G6 and Q7) on the flat plane surface of CBM1, Cel7A was found to prefer to interact with hardwood lignin (*Eucalyptus globulus*) compared to softwood *Cryptomeria japonica*. This is in line with our results.

Proteins with molecular weights above 70 kDa were found to be adsorbed at significantly higher rates (30–100%) than CBHs and CD (1–15%). In particular, 80–90 kDa proteins were found to be adsorbed onto rice lignin at rates of 97.7–99.5%. The sizes of secreted proteins of *T. reesei*, namely swollenin (80 kDa), β-glucosidase (Cel3A, 81 kDa), and endoglucanase (Cel74A, 87.1 kDa) correspond to the size of the observed bands [36]. Hydrophobic patch scores on the enzyme structure also correlate with enzyme–lignin adsorption [9]. Accordingly, β-glucosidase (Bgl1) of *A. niger* has the highest hydrophobic patch score (45.9) compared to enzymes such as Cel7A (13.3), acetyl xylan esterase (9.1), and endoxylanase (0.8) of *T. reesei*. No scores of a hydrophobic patch on β-glucosidase of *T. reesei* have been obtained. However, the β-glucosidase of *A. niger* was found to exhibit less adsorption onto lignin than β-glucosidase of *T. reesei* [41]. The extracellular β-glucosidase, Cel3A, accounts for 1.38% of total secreted proteins [36] and displays high adsorption rates onto the different types of lignin (Figure 2). These results thus indicate that the bands shown in Figure 2 correspond to β-glucosidase Cel3A (81 kDa). Two proteins with approximately 20 and 23 kDa molecular weights were found to be adsorbed at higher rates onto lignin of woody plants (oak and pine) than that of herbaceous plants (rice and kenaf). These may correspond to xylanases XYNI and XYNII. Proteins with lower molecular weights ranging from 6 to 10 kDa observed with kenaf lignin were also found to be adsorbed at 100%. Further studies are required to more clearly identify these proteins.

### 3.3. Adsorption of Endoglucanases onto Lignin

Endoglucanases (Endo-1,4- β-D-glucanases, EGs) randomly cleave internal β-1,4-glucoside bonds in the cellulose chain, and account for 8.46–17.4% of the secreted proteins of *T. reesei*, which were found to be composed of 5–10% Cel7B, 1.26–2.40% Cel7B, 2% Cel5A, 0.05–1.96% Cel12A, and 0.18–0.78% Cel61A [36,42]. The activity of EGs accelerates the saccharification rate with CBHs, and due to generation of more reducing and non-reducing ends of cellulose chains.

The EGs produced by *T. reesei* have been classified into two glycoside hydrolysis family 5 proteins (GH 5), Cel5A (48 kDa) and Cel5B (46.8 kDa), a GH 7 protein Cel7B (55 kDa), a GH 12 protein Cel12A (26 kDa), a GH 45 protein (Cel45A, 24.4 kDa), two GH 61 proteins, Cel61A (35.5 kDa) and Cel6B (26 kDa), a GH 74 protein Cel74A (87.1 kDa), and an endoglucanase candidate enzyme (36 kDa) [36]. We observed activities of EGs on CMC substrate (these EGs are referred to as the enzymes that are active on CMC) in SDS-PAGE and zymogram assay results (Figure 3A). These activities are manifested on bands including proteins such as E1 (~100 kDa), E2 (Cel74A, 87 kDa), E3 (Cel7B, 55 kDa), E4 (Cel5A, 48 kDa), E5 (~40 kDa), E6 (~30 kDa), E7 (~20 kDa), E8 (~15 kDa), and E9 (~10 kDa). The ratio of each enzyme in total endoglucanase content was determined by analyzing band intensities on the gel. E3 and E4 were found to account for 48.01% and 24.61% of the total EG activity, respectively. By comparison, E1, E2, E5, E6, E7, E8, and E9 were found to be responsible for 7.45%, 3.13%, 5.33%, 2.60%, 2.18%, 4.63%, and 2.06% of the total activity, respectively (Figure 3A).

The EG adsorption onto lignin of oak, pine, rice, and kenaf was determined using the following equation: [EGs adsorption] = [Total EGs activity] − [Free EGs activity in supernatant after lignin adsorption]. Surprisingly, EG activity was found to be elevated by 10~25% after adsorption onto lignin (Figure 3B). EG activity on CMC in the cellulase cocktail is connected to CBH and β-glucosidase, and is directly dependent on β-glucosidase to accelerate the hydrolysis rate of the substrate [43]. The underlying reason for this increased activity of EGs after adsorption onto lignin is discussed in the following section with β-glucosidase. Zymogram assay is useful to measure adsorption of specific endoglucanase types with different binding affinities to lignin without CBHs and β-glucosidase activity. Zymogram results are shown in Figure 3C–G. The major endoglucanase, E3, was found to be adsorbed less than CBHs with similar molecular weight and cellulose binding motif 1 family (CBM1), and showed two times higher adsorption in lignin of the herbaceous plants rice and kenaf. Enzymes E1 and E2 with high molecular weights were found to be adsorbed onto lignin at rates of 2.15–3.1% and 1.5–6.29%, respectively. Adsorption of E2 was also found to be 2–4 times higher compared to that of herbaceous plants. Finally, adsorption of E6 was found to be 2 times higher onto oak lignin compared to other lignin types.

### 3.4. Adsorption on β-Glucosidases onto Lignin

Ten β-glucosidase isozyme (Bgls) genes of *T. reesei* exist according to the *T. reesei* genome database v.2.0 and Carbohydrate-Active Enzymes (CAZy) database. The Bgls include two isozymes (Cel1A and Cel1B) of the glycoside hydrolase family 1 (GH 1) and eight in the GH 3 family (Cel3A, B, C, D, E, F, G, H) [44]. The Bgls can be also classified according to the protein destinations into two groups: extracellular Bgls including Cel3A (81 kDa), Cel3B (108 kDa), Cel3F (139 kDa), and Cel3G (129 kDa); and intracellular Bgls including Cel1A (BglII, 55 kDa), Cel1B (58 kDa), CelC (100 kDa), Cel3D (90 kDa), and Cel3E (104 kDa). Among these, Cel3A has been quantified in the secretome of *T. reesei* and was found to account for 1.38% of total secreted proteins [33]. Cel3A was also shown to have high affinities for cellotriose, sophorose, and laminaribiose. Cel3B has the highest specific activity on cellobiose, followed by Cel1A and Cel3A. Cel1A of the Bgls was found to have the highest affinity for cellobiose and tolerance to the end product glucose [44]. This is advantageous to enhance the conversion rate of cellobiose to glucose by mutagenesis against end-product inhibition [45].

End-product inhibition is known to cause reduction in the enzymatic hydrolysis rate. Cellobiose is a particularly strong inhibitor of Cel7A, and leads to the loss of 50% of the enzyme activity at 19 mmol L^−1^ cellobiose concentration, whereas Cel6A loses 25% of its activity at 42 mmol L^−1^ cellobiose concentration [46]. The β-glucosidase of *T. reesei* loses its activity upon adsorption onto lignin isolated from liquid hot water-pretreated hardwoods, whereas the activity is maintained in the case of Bgls of *A. niger* [41]. This is considered to be due to the adsorption of β-glucosidase onto lignin, leading to the increase in cellobiose concentration in the early stages of enzymatic hydrolysis and resulting in the severe reduction in the hydrolysis rate. Supplementation with β-glucosidase of *A. niger* avoids the reduction in the hydrolysis rate due to lignin adsorption.

An adsorption experiment of β-glucosidases in the cellulase of *T. reesei* onto lignin was performed at 50 °C for 1 h in 1 mL of 20 mM citrate buffer including 10 mg mL^−1^ lignin. The activities of free enzymes in the supernatants after adsorption onto lignin were measured with substrates *p*NPG or cellobiose. The β-glucosidases were found to be adsorbed at the highest rates onto kenaf lignin (Figure 4). However, when glucose yields are compared, different rates of adsorptions of β-glucosidase isozymes are observed. Here, the highest conversion rate of cellobiose to glucose was shown in the fraction of rice and kenaf lignin adsorption, although the lowest cellulose consumption was shown, which indicated that the β-glucosidase isozyme possessing the transglycosylation function was dominantly adsorbed by rice and kenaf lignin. Previous reports indicated that Cel3A has a higher K_cat_ (S^−1^) value on cellobiose, and higher transglycosylation activity on cellobiose to cellotriose than those of Cel3B [44]. The higher hydrophobicity and pI point of cellulolytic enzymes contribute to higher lignin adsorption [9,30,38]. Cel3A has also been characterized to have a higher isoelectric point (8.5) and hydrophobicity value (−0.163) than those of Cel3B (5.73 and −0.317, respectively) [41]. These results indicate that Cel3A was dominantly adsorbed onto the lignin of rice and kenaf under a pH 5.0 condition and resulted in lower cellulose consumption and a higher glucose yield in the fraction of the herbaceous lignin. This result also explains why, in Figure 4B, EG activities after lignin adsorption are increased.

*Aspergillus niger* is one of the most efficient producers of β-glucosidase. To date, 17 β-glucosidase encoding genes have been identified in the *A. niger* genome [47], and even more β-glucosidase genes have been estimated to exist (JGI MycoCom, https://mycocosm.jgi.doe.gov/Aspni7/Aspni7.home.html, accessed on 1 December 2021). These enzymes can be classified to belong to the GH family 1 and/or GH 3 [48]. Eight enzymes were found to be secreted into the extracellular environment [47]. β-glucosidases also have two times higher hydrolytic activity on cellobiose than on *p*-NPG according to the supplier’s reference and Seidle et al. (2004) [49].

Adsorption of *A. niger* β-glucosidase onto lignin was performed at 50 °C for 1 h in 1 mL of 20 mM citrate buffer (pH 5.0) including 10 mg mL^−1^ lignin. β-glucosidases adsorbed onto lignin were separated via SDS-PAGE (Figure 4C–E). The band B4 (approximately 56 kDa) included enzymes highly adsorbed onto the lignin structure at rates of 52–130%. Two β-glucosidases of *A. niger* belonging to the GH family 1 and relatively small molecular weight (54.2 and 64 kDa) have been reported previously [47,50]. The genes encoding two β-glucosidases (Accession No: An03g03740 and An11g02100) were predicted to be secreted into the extracellular environment (BUSCA: http://busca.biocomp.unibo.it/, accessed on 1 December 2021); however, An03g03740 (designated Bgl1B, 54.2 kDa) was also confirmed to be an intracellular protein in vivo in *Saccharomyces cerevisiae* [51]. Here, B4, an extracellular β-glucosidase belonging to the GH family 1, is highly adsorbed onto lignin. Regarding β-glucosidases belonging to GH family 3 with higher molecular weights, of between 95 and 170 kDa (B2 and B3), B2 was found to be adsorbed at 15–20% onto all lignin types except for kenaf lignin (3%), and B3 was found to adsorbed onto lignin 0.5–4.2 times higher than the control sample. β-glucosidase (Bgl1, 117 kDa) of *A. niger* was also found to have the highest hydrophobicity patch score compared to serum albumin, Cel7A (cellobiohydrolase 1), Axe1 (acetyl xylan esterase), E1 (endoglucanase), XynA (endoxylanase), AbfB (arabinofuranosidase), and Xyn11 (endoxylanase) [9]. This result explains why B3 (110 kDa) was found to be most adsorbed onto the lignin.

### 3.5. Xylanase Adsorption by the Lignins

Hemicelluloses are heterogeneous polysaccharides composed of different combinations of sugar polymers such as xylans, arabinans, mannans, and galactans, depending on the source and type of lignocellulosic biomass [52]. The hemicellulose of hardwood is composed of 80–90% *O*-acetyl-4-*O*-methylglucuronoxylan, 0.1–1% arabinomethylglucurononxylan, and 1–5% glucomannans. The main component of softwood hemicellulose is galactoglucomannan, which accounts for 60–70% of hemicellulose. Galactoglucomannan is followed by arabinomethylglucuronoxylan and methyglucuronoxylan, which account for 13–30% and 5–15% of the hemicellulose, respectively.

β-xylanases (Endo-1,4- β-D-xylanases) randomly cleave internal β-1,4-xylosidic bonds within the xylan structure, and account for 0.42% of the secreted proteins of *T. reesei* [36]. The composition of secreted β-xylanases was found to be composed of 0.25% XYNI (19~21 kDa), 0.07% XYNII (20~21 kDa), XYNIII (not measured, 38 kDa), 0.10% XYNIV (55 kDa), XYNV (not measured, 21 kDa), and XYNVI (not measured, 57 kDa) [35,36,37,53]. Two xylanases (XYNI and XYNII) from the GH family 11 are known to be major xylanolytic enzymes of *T. reesei*. However, in this study we found the GH family 30 XYNIV was the main enzyme of xylanases in the cellulase cocktail of *T. reesei* in terms of activity on the soluble beechwood xylan (Figure 5).

Xylan derivatives formed in the early stages of enzymatic hydrolysis, especially xylooligomers, are strong inhibitors of cellulases Cel7A and Cel6A [54]. Guo et al. (2014) previously reported that the crude xylanase produced by *Penicillium* sp. was most adsorbed onto pine lignin and showed about 45% of inhibition. The adsorption ratios of different lignin types were arranged in the following decreasing order: pine > corn stove > aspen > kenaf [18]. Therefore, the reduction in endo-β-xylanase and β-xylosidase activities due to adsorption onto lignin is likely a severe problem hindering rapid and economic enzymatic saccharification for bioconversion.

Activities of free xylanases in the control cellulase cocktail of *T. reesei* and in the supernatant obtained after centrifugation were measured (Figure 5A). The activities of free xylanases in the supernatants were unexpectedly increased for all lignin types. To confirm adsorption of xylanases onto lignin without enzyme dissociation, samples were incubated with soluble beechwood xylan, and xylanase activities were measured. The ratio of the activities of the adsorbed xylanase were found to be 6.5–8.5% with respect to the control. The free xylanase activity in the supernatant is due to the presence of β-xylosidase and various xylanolytic enzymes which correspond to several bands at molecular weights of approximately 110 kDa (X1), 100 kDa (X2), and 80 kDa (X3) in Figure 5B. In addition, small molecular weight XYNI and XYNII have been known to be the main enzymes in terms of expression levels, and not detectable under the zymogram experimental conditions. The same result was also obtained for free xylanase activity after adsorption onto lignin for xylanase from *T. lanuginosus* (Figure 5D). This is not representative of adsorption of endo β-xylanases onto lignin. Zymogram activity results show the adsorption of a major xylanase, XYNIV, in Figure 5B,C. Accordingly, XYNIV was found to be adsorbed onto lignin at rates of 1-2%, and kenaf lignin displayed the lowest binding affinity to the enzyme. This is consistent with the results of Guo et al. (2014) [18]. Lignin from woody plants were also found to allow adsorption of more XYNIV than those of herbaceous plants. X1 with a higher molecular weight was found to be adsorbed onto pine lignin more than oak lignin, and kenaf lignin yielded higher adsorption than rice straw lignin.

The xylanase of *T. lanuginosus* xylanase was used to estimate adsorption rates of xylanases with small molecular weights. Hemicellulase produced by *T. lanuginosus* is composed of β-xylanase (Xyn11A, GH 11, 24.3 kDa) and β-xylosidase (GH 43, 38.1 kDa) [55,56]. The β-xylanase is expressed at high quantities and shows high thermal stability; therefore, it is commercially useful. The small molecular weight xylanases, XYNI and XYNII, belonging to GH family 11 in the cellulase cocktail of *T. reesei*, have low activities on beechwood xylan, which renders measurement of the enzyme–lignin affinity difficult. The xylanase Xyn11A, from *T. lanuginosus* belonging to the same GH family and highly active on beechwood, can be surrogated for the XYNI and XYNII of *T. reesei.*

Adsorption of Xyn11A onto lignin was conducted, and supernatant and lignin pellet fractions were obtained. The xylanases on the different types of lignins were dissociated from lignin and loaded onto SDS-PAGE gels containing beechwood xylan for zymogram analysis. The xylanase with the molecular weight of approximately 23 kDa was found to be adsorbed by up to 17.0% onto oak lignin. Oak lignin was followed by pine, rice straw, and kenaf lignin showing 13.4%, 11.6%, and 9.0% adsorption onto lignin, respectively (Figure 5D,F). The single xylanase (Xyn11A) adsorption onto the lignin was shown to be 6.6–8.9 times higher than that of XYNIV in the cellulase cocktail of *T. reesei*. When the Xyn11A of *T. lanuginosus* is used to supplement the cellulase cocktail, the quantity of the enzyme that is adsorbed can be determined. 

The free *T. reesei* and *T. lanuginosus* xylanase activities in the supernatants were observed to increase by 17–35% and 10–20% compared to the control, respectively. This is distinguishable from the different levels of increase between the xylanase activities of *T. reesei* and *T. lanuginosus* depending on the complexity of the enzyme components. There are many cellulolytic and helping enzymes containing carbohydrate binding modules (CBM1 is dominant) in the cellulase cocktail of *T. reesei* [53]. It cannot be ruled out that these enzymes bind other polysaccharides. Type B CBMs are known to be able to interact with cellulose and xylan substrates, and the CBM of endoglucanase of microorganisms in buffalo rumen can also bind diverse substrate polymers such as Avicel, birchwood xylan, mannan, lichenan, and raw starch [57]. The increase in the *T. reesei* xylanase activity after adsorption onto lignin may thus be considered to be induced by a reduction in the amounts of competitors such as the endoglucanases and helping enzymes containing CBMs. Moreover, the activity of the xylanase of *T. lanuginosus* may also have been induced by reduction in the masking effect on the substrate due to adsorption onto lignin [7]. This result and the adsorption rate of the xylanases are available to prepare a cost-efficient hydrolytic enzyme cocktail.

### 3.6. Adsorption of Mannanase onto Lignin

Endo-1,4- β-D-mannanase cleaves internal β-1,4-D-glycosidic bonds of the mannan backbone chain, and several enzymes, such as β-mannosidase, β-glucosidase, acetyl-mannan esterase, and α-galactosidase, participate in the hydrolysis of galactoglucomannan in the hemicellulose of softwood [58]. β-mannanase was also reported to act synergistically with xylanase during the enzymatic hydrolysis of softwood [59,60,61].

A mannanase produced by *T. reesei* has been previously identified as Man5A (experimentally 53.6 kDa) or mannanase I (MANI, experimentally 53 kDa, accounting for 0.25% of the secreted proteins) containing CBM1 [36,62]. Here, a small amount of Man5A in the cellulase cocktail was adsorbed onto the woody lignin of oak and pine at rates of 1.21% and 1.74%, and at 1.83% and 2.50% in the herbaceous lignin of rice and kenaf, respectively (Figure 6). The supplement including recombinant mannanase remarkably accelerated the saccharification rate of softwood [60], which implies that adsorption of mannanase onto lignin shows no synergistic effect in the early stages of hydrolysis, and results in severe retardation of saccharification.

### 3.7. Summary of Lignin Adsorption of Major Enzymes in the Cellulase Cocktail of T. reesei

A comparative evaluation of adsorption of specific enzymatic species in the cellulase cocktail onto lignin was performed to analyze their levels of interaction with different types of lignin (Table 2). The absorption of the full-length CBHs (CD+CBM) was 97% and 43% higher onto the oak and kenaf lignin than the rice and pine lignin. The adsorption ratio indicates that the enzyme has higher affinity to the lignin having a higher S/G ratio. Adsorption of CD was in the reverse order. The CBM1 of CBHs with high hydrophobicity contributed to higher binding affinities of oak and kenaf lignins. The Cel7B was adsorbed at higher levels onto lignins of herbaceous plants, whereas XYNIV was adsorbed at higher levels onto the lignin of woody plants. The β-glucosidase, Cel3A, was adsorbed onto rice lignin at high levels (approximately 95%), at 51–57% onto the woody plant lignin, and at 19% onto the kenaf lignin. The high hydrophobicity of Cel3A also caused remarkable levels of adsorption onto lignin compared to other enzymes. Finally, kenaf lignin yielded the highest adsorption of mannanase.

The proteins of the cellulase cocktail of *T. reesei* were adsorbed by 25.3%, 19.6%, 17.1%, and 21.8% of the total secreted proteins by 10 mg mL^−1^ lignin of oak, pine, rice straw, and kenaf, respectively. The major enzymes (CBHs, Cel7B, Cel3A, XYNIV, and Man5A) were adsorbed by 9.09%, 6.67%, 7.61%, and 7.12% (Table 2). The others, which contain small amounts of enzymes such as non-quantified β-glucosidases, endoglucosidases, xylanases, β-xylosidases, and accessory enzymes (polysaccharide monooxygenases, esterases, arabinofuranosidase, swollenins, etc.) were more adsorbed, by 16.2%, 12.9%, 9.5%, and 14.7% by the lignin of oak, pine, rice straw, and kenaf, respectively, which is inferred to be responsible for the enzymatic retardation at the early stages of saccharification.

## 4. Conclusions

Adsorption of enzymes onto lignin during processing of lignocellulosic biomass leads to severe problems and increases the cost of the bioconversion process. Major cellulolytic and xylanolytic enzymes are CBHs, Cel7B, Cel3A, XYNIV, and Man5A. Here, we quantified the adsorption of these enzymes onto lignin under various practical enzymatic hydrolysis conditions. Total adsorption ratios of the major enzymes onto different lignin types from oak, pine, rice straw, and kenaf were found to be 9.09%, 6.67%, 7.61%, and 7.12%, respectively, with respect to the 82.13% secretome of *T. reesei*. Adsorption of hydrolytic enzymes onto lignin, especially that of helping enzymes such as β-glucosidase, xylanase, and mannanase, is inferred to render enzymatic hydrolysis inefficient in the early stage. In addition, the adsorption on low-expressed secretome enzymes in the cellulase cocktail, of 9.5–16.2%, is predicted to impede the acceleration of the enzymatic hydrolysis rate. The enzymatic and physical interference of lignin hinders the economic potential of the pretreatment and saccharification process. For this purpose, delignification of lignocellulosic biomass via pretreatment can help achieve economic saccharification processes.

## Figures and Tables

**Figure 1 polymers-14-00167-f001:**
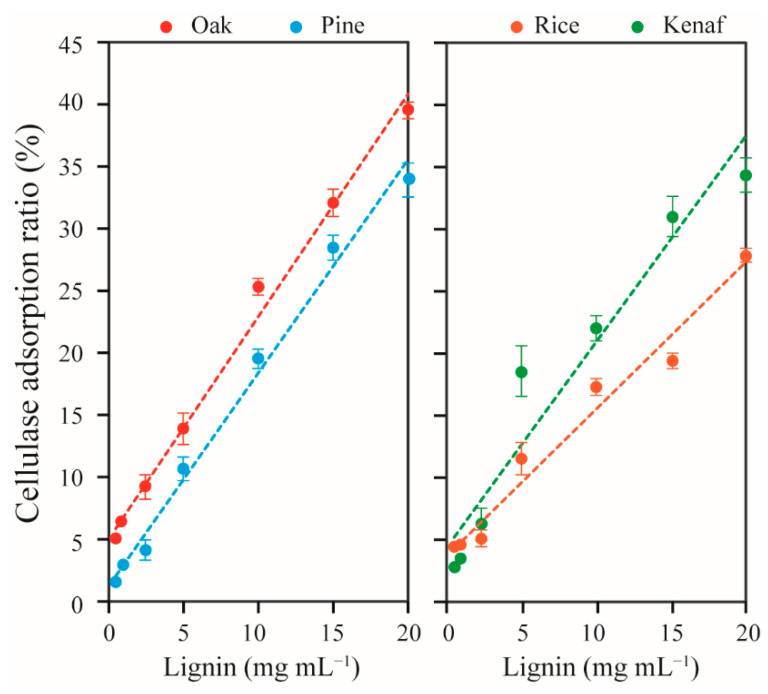
Rates of adsorption of extracellular enzyme onto lignin. Lignin isolated from oak, pine, rice straw, and kenaf were incubated in 1 mL of 20 mM citrate buffer (pH 5.0) with cellulase cocktail of *T. reesei*.

**Figure 2 polymers-14-00167-f002:**
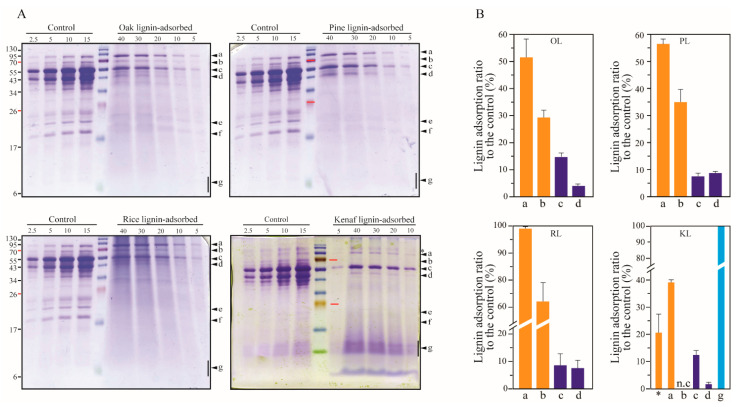
Adsorption of CBHs and high molecule weight protein in the cellulase of *T. reesei* onto lignin. (**A**) The proteins dissociated from 10 mg mL^−1^ lignins were separated via SDS-PAGE and stained with Coomassie blue and (**B**) the bands were quantified by analyzing band intensities. The numbers on the upper side of the gel indicate loading volumes (μL). *, 95 kDa (predicted to β-xylosidase or β-xyloglucanase); a, 81 kDa (Cel3A: β-glucosidase, *p* = 0.010); b, 70 kDa (*p* = 0.333); c, 55 kDa (Cel7A and Cel6A, *p* = 0.004); d, 45 kDa (CD: Catalytic domain of CBHs, *p* ≤ 0.001); e and f, xylanases (XYNI and XYNII); g, low molecular weight proteins (6~10 kDa); n.c, not calculated. OL, oak lignin; PL, pine lignin, RL, rice straw lignin; KL, kenaf lignin.

**Figure 3 polymers-14-00167-f003:**
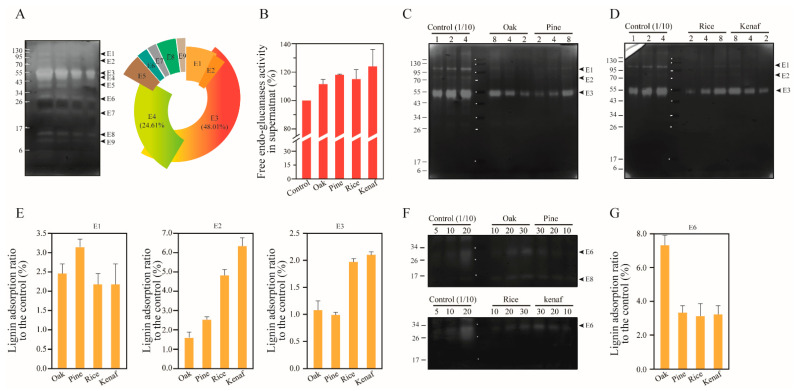
Adsorption of endoglucanases (EGs) in cellulase cocktail of *T. reesei* onto lignin. (**A**) Endoglucanases in the cellulase were quantified by measuring their activities on CMC substrate in SDS-PAGE. The activity ratios were as follows: E1 (7.45%), E2 (3.13%), E3 (48.01%), E4 (24.61%), E5 (5.33%), E6 (2.60%), E7 (2.18%), E8 (4.68%), and E9 (2.06%). (**B**) Adsorption of EGs onto lignin was measured by comparing the activities of EGs in the supernatants to that of the control following lignin–enzyme precipitation (*p* = 0.121). (**C**–**G**) Adsorption of EGs onto lignin was analyzed by zymogram assay using CMC substrate in SDS-PAGE, for lignin of the woody plants oak and pine (**C**) and the herbaceous plants rice straw and kenaf (**D**). The band intensity was analyzed for quantification and comparison of the adsorptions of EGs (**E**,**G**). The zymogram activity of E6 was shown (**F**). The numbers on the upper side of the gel indicate loading volumes (μL). E1, ~100 kDa (*p* = 0.070); E2, ~87 kDa (Cel74A, *p* = 0.022); E3, 55 kDa (Cel7B, *p* = 0.006); E4, 48 kDa (Cel5A); E5, ~40 kDa; E6, ~30 kDa (*p* = 0.004); E7, ~20 kDa; E8, ~15 kDa; E9, ~10 kDa.

**Figure 4 polymers-14-00167-f004:**
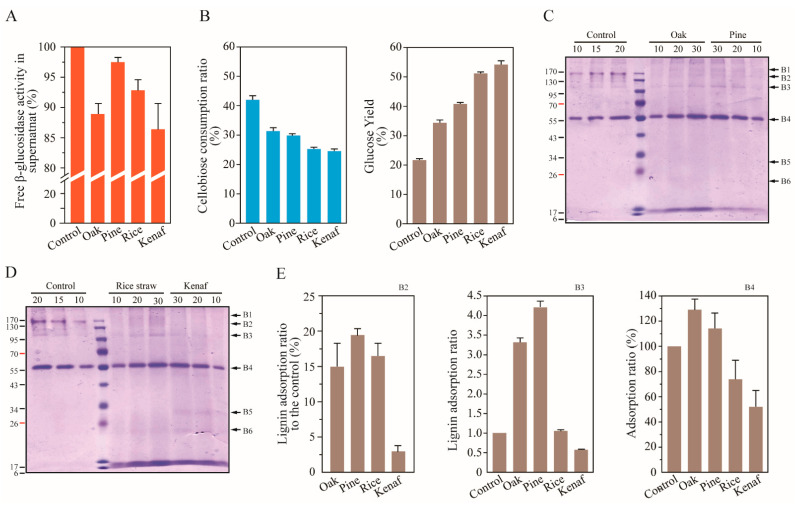
β-glucosidase inhibition upon adsorption onto lignin. β-glucosidases activities in *T. reesei* cellulase cocktail were measured via quantification of *p*NPG (**A**) and cellobiose (**B**), and adsorption ratio was analyzed (*p* = 0.033). (**B**) Transglycosylation activity of β-glucosidases were observed to be high in the control, based on the observed correlation between cellobiose consumption (*p* ≤ 0.001) and glucose production (*p* ≤ 0.001). (**C**–**E**) Adsorption of *A. niger* β-glucosidase onto lignin was performed, and the enzymes dissociated from the lignin were loaded onto the SDS-PAGE gel (**C**,**D**), and band intensities were analyzed (**E**). B1 concentration was increased due to adsorption onto lignin, especially for oak and pine lignin. B2 and B3 corresponded to major bands of β-glucosidase belonging to GH family 3. The numbers on the upper side of the gel indicate loading volumes (μL). B1: 190 kDa; B2: 160 kDa (*p* ≤ 0.001); B3: 110 kDa (*p* = 0.005); B4: 56 kDa (*p* ≤ 0.001); B5: 30 kDa; B6: 20 kDa.

**Figure 5 polymers-14-00167-f005:**
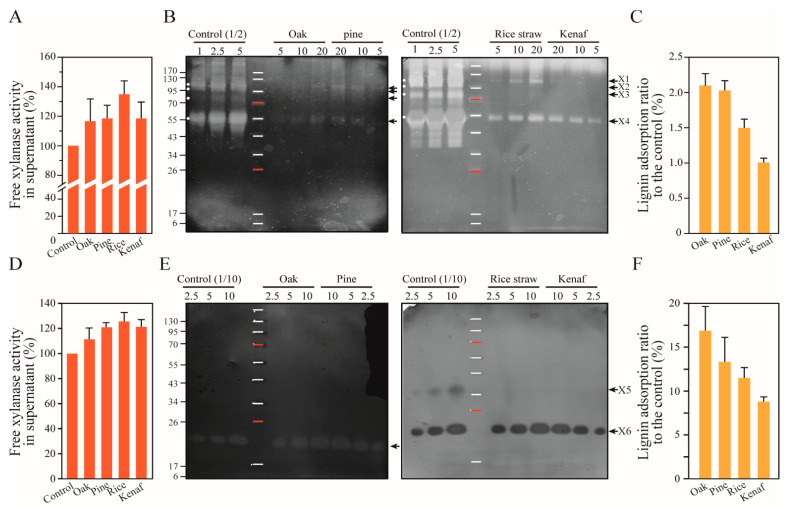
Adsorption of *T. reesei* and *T. lanuginosus* xylanases onto lignin. (**A**) Adsorption of xylanase in the cellulase cocktail of *T. reesei* onto lignin was determined by measuring the xylanase activity in the supernatant (*p* = 0.361) and (**B**,**C**) zymogram activity in the lignin pellet fraction on soluble beechwood xylan. Several bands indicating xylanase activity were observed on the gel at positions corresponding to approximately 110 kDa (X1), 100 kDa (X2), and 80 kDa (X3). The band intensity of the major xylanase, XYNIV (X4, 55 kDa) was determined (*p* = 0.001). Woody lignin of oak and pine were found to yield higher adsorption than herbaceous lignin of rice and kenaf. (**D**–**F**) Adsorption of *T. lanuginosus* onto lignin. The free xylanases in the supernatant were measured (*p* ≤ 0.001). The bands indicating xylanase (23 kDa, X6) were visible on SDS-PAGE gel containing beechwood xylan. Intensities of these bands were determined to quantify adsorption of the enzyme (*p* = 0.037). β-xylosidase (X5) was shown at the position of molecular weight 38 kDa in the control sample. The numbers on the upper side of the gel indicate loading volumes (μL).

**Figure 6 polymers-14-00167-f006:**
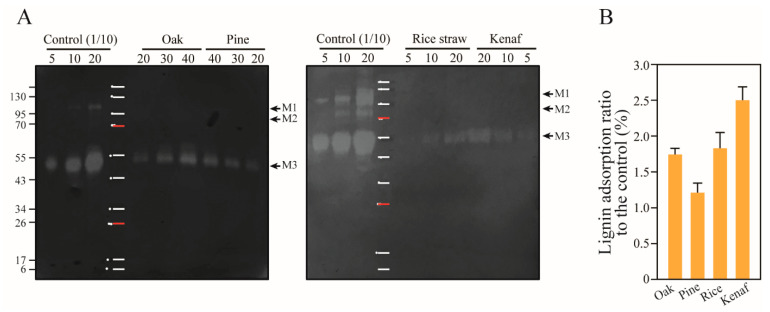
Adsorption of *T. reesei* mannanase onto lignin (**A**) Zymogram activity of mannanase adsorbed onto the lignin of oak, pine, rice straw and kenaf was assessed. The bands on SDS-PAGE containing glucomannan were observed at the 53 kDa position (M3), which indicates that Man5A is the major mannanase. (**B**) The intensities of the bands (M3) were determined to compare ratios of adsorption with different types of lignin (*p* = 0.409). The numbers at the top of the gels represent the loading volume (μL). The control was diluted 10×. The two bands showing mannanase activity were observed at the positions corresponding to approximately 110 and 80 kDa, which are indicated with M1 and M2, respectively.

**Table 2 polymers-14-00167-t002:** Rates of adsorption of the major enzymes in the cellulase cocktail produced by *T. reesei* onto different types of lignin.

	Molecular Weight (kDa)	Lignin Adsorption Rate (%)			
		Oak	Pine	Rice Straw	Kenaf
Cellobiohydrolase					
CD ^1^ + CBM ^2^	55 kDa	14.61 ± 1.94	7.40 ± 1.63	8.74 ± 3.27	12.52 ± 2.69
CD	48 kDa	4.00 ± 0.84	8.77 ± 0.42	7.73 ± 1.51	1.63 ± 0.17
CD/CD+CBM		0.27	1.19	0.88	0.14
Endoglucanase					
Cel7B	55 kDa	1.21 ± 0.14	1.02 ± 0.05	1.96 ± 0.34	2.09 ± 0.37
ꞵ-glucosidase					
Cel3A	81 kDa	51.66 ± 8.66	56.71 ± 2.08	94.68 ± 5.91	19.24 ± 5.03
Xylanase					
XYNIV	55 kDa	2.10 ± 0.26	2.03 ± 0.21	1.50 ± 0.17	1.01 ± 0.06
Mannanase					
Man5A	53 kDa	1.74 ± 0.71	1.21 ± 0.80	1.83 ± 0.36	2.50 ± 0.57
Sum ^3^		9.09	6.67	7.61	7.12

^1^ CD, catalytic domain. ^2^ CBM, carbohydrate binding module. ^3^ The sum was calculated with the ratio of the enzyme amount to the total amount of cellulase of *T. reesei*. The enzyme amount ratios are presented such as CD + CBM (50.54%), CD (22.46%), Cel7B (7.5%), Cel3A (1.38%), XYNIV (0.1%), and Man5A (0.25%), based on Herpoel-Gimbert et al. 2008 [36]. This corresponds to 82.13% of the secretome protein of *T. reesei*.

## Data Availability

The data presented in this study are available on request from the corresponding author.

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
