# Peer review of "Comparative Evaluation of Adsorption of Major Enzymes in a Cellulase Cocktail Obtained from Trichoderma reesei onto Different Types of Lignin"

_polymers, 2022, doi:10.3390/polym14010167_

Round 1

Reviewer 1 Report

Reviewer’s report

The manuscript entitled: “Comparative evaluation of adsorption of major enzymes in cellulase cocktail obtained from Trichoderma reesei onto different types of the lignin”. The Authors focused on the research the adsorption role of lignin.

Comments and Suggestions for Authors

Dear Authors,

In my opinion, the article requires a major revision.  The main problem with the authors' study findings is the lack of statistical analysis performed. In the following document part, Authors often use the term “correlation” between measured quantities. There is no introduction of the concept of what means for example "high correlation" and how it was calculated?

Detail comments:

Line 99:  Some language editing is needed, in line 99 the following sentence is not clear “…to one atmospheric pressure.” The reactor was opened and the sample was exposed to atmospheric pressure, but did Authors measure the pressure at that moment? It is very simplicity tell that atmospheric pressure value is strictly “an one”.

Line  98-99: I suggest to change the pressure units to Pa or MPa. By the way, it is important to highlight in the Methods part of the manuscript to strongly write that pressure is a relative pressure to atmospheric pressure, or it is absolute pressure. Usually, it would also be useful to specify the type and accuracy of the gas pressure gauge used in the experiment.

Line 110-111:  It is worth providing the acceleration or the model of the centrifuge used in the experiment. In my opinion, the information about the rotational speed is insufficient, because it says nothing about the accelerations and forces affecting the material placed in the centrifuge.

Line 132-134: Estimation of adsorption rate by use the Adobe PhotoShop software histogram need clarify or input appropriate citations.  What kind of picture layer has been used, which canal etc.

Line 231-240:  In this section can be benefited by using tables and graphs to help your description. I suggest add the graph contain of G, H, S  lignin subunits as a function of origin source of lignin: hardwood, softwood or kind of plant.

Line 499-502: “The xylanase with the molecular weight of approximately 23 kDa was found to be adsorbed up to 17.0% with respect to the control onto oak lignin ….” . Why that situation have been observed , does the Authors can any idea to explain it?

Line 553-556: In the sentence: “The adsorption ratio was found to be correlated to the S/G ratios, which were 1.50-1.84 and 1.40-1.8 in oak and kenaf lignin, and 0.34 and 0.0043 in rice and pine lignin, respectively.”. How the correlation between adropion and S/G ratio was found. How much was the correlation coefficient? 

Did the Authors conduct any statistical analysis during the study?

Reviewer 2 Report

The authors have elucidated the binding of different hydrolytic enzymes produced by T. reesei to the lignins isolated from woody and herbaceous species. They utilized basic techniques like colorimetry, gel separation, and gel color intensity measurement to elucidate enzyme binding. Although the findings are not novel and that the analytical techniques are not comprehensive, the authors have exemplified their study by conducting exhaustive experiments, analyzing different hydrolytic T. reesei enzymes. This research provides additive information which may benefit other related works. Hence, this manuscript may be published provided the following minor corrections are incorporated.

1) Please check the unit (of 21 kg/cm) in Line 99. Is this the unit of pressure, please verify?

2) What are the dimensions and/or capacity of the pretreatment reactor?

3) Figure 1 caption: No need to mention that it was Celluclast brand (or the 1.5 L capacity), again and again. That the brand name was mentioned in the materials section is sufficient.

4) Figure 2 caption, Line 308: Is the unit of gel loadings in microliter? If so, please mention in the figure labels or in the caption within parenthesis.

5) What is the unit of rates of adsorption provided in Table 1? If it is in percentages, please cite.

6) Provide abbreviation for CD and CBM in Table 1's footnotes.

7) Conclusion: Not sure if the binding of 7 to 9% enzymes could cause severe retardation of saccharification rates. In fact, studies as early as 2008 have shown that the addition of bovine serum albumin could easily prevent non-catalytic enzyme binding. Hence, the study results ONLY indicate a potential increase in saccharification costs (which is mentioned in Line 575), and not necessarily a severe loss of saccharification efficiency.

8) Please double check the spelling and grammar throughout the manuscript. Few corrections, but not all, are provided below:

(a) Line 24: Our results "showed" that..... 

(b) Line 25: lignin types affect (not effect)...

(c) Line 128: brought to "boiling" (not boing)...

(d) Line 238: corn stover...... (not corn stove)

(e) Line 258: Lignin isolated from oak,.....

(f) Line 542: different types of lignin

Round 2

Reviewer 1 Report

Dear Authors, thank you for improving the manuscript. I think that now your research report is ready for publication. 

Best regards

Reviewer